# Structural insights into functional properties of the oxidized form of cytochrome *c* oxidase

Izumi Ishigami [1], Raymond G. Sierra [2], Zhen Su[2,3], Ariana Peck[2], Cong Wang[2], Frederic Poitevin[2], Stella Lisova[2], Brandon Hayes[2], Frank R. Moss III[2,4], Sébastien Boutet [2], Robert E. Sublett[2], Chun Hong Yoon [2], Syun-Ru Yeh [1] ✉ & Denis L. Rousseau [1] ✉

Cytochrome *c* oxidase (C*c*O) is an essential enzyme in mitochondrial and bacterial respiration. It catalyzes the four-electron reduction of molecular oxygen to water and harnesses the chemical energy to translocate four protons across biological membranes. The turnover of the C*c*O reaction involves an oxidative phase, in which the reduced enzyme (R) is oxidized to the metastable $O_H$ state, and a reductive phase, in which $O_H$ is reduced back to the R state. During each phase, two protons are translocated across the membrane. However, if $O_H$ is allowed to relax to the resting oxidized state (O), a redox equivalent to $O_H$, its subsequent reduction to R is incapable of driving proton translocation. Here, with resonance Raman spectroscopy and serial femtosecond X-ray crystallography (SFX), we show that the heme $a_3$ iron and $Cu_B$ in the active site of the O state, like those in the $O_H$ state, are coordinated by a hydroxide ion and a water molecule, respectively. However, Y244, critical for the oxygen reduction chemistry, is in the neutral protonated form, which distinguishes O from $O_H$, where Y244 is in the deprotonated tyrosinate form. These structural characteristics of O provide insights into the proton translocation mechanism of C*c*O.

Mammalian C*c*O is a large integral membrane protein comprised of 13 subunits. It contains four redox active centers, $Cu_A$, heme *a*, and a heme $a_3$/$Cu_B$ binuclear center (BNC) (Fig. 1A). Molecular oxygen binds to the heme $a_3$ iron in the BNC, where it is reduced to two water molecules by accepting four electrons from cytochrome *c* and four protons (the "substrate" protons) from the negative side (N-side) of the mitochondrial membrane. The energy derived from the oxygen reduction chemistry is used to drive the translocation of four protons (the "pumped" protons) from the N-side to the positive side (P-side) of the membrane[1,2]. Strong evidence suggests that the substrate protons are delivered to the BNC via the D and K-channel (see Supplementary Fig. 1), while the pumped protons are translocated via the D-channel[3–5] or the H-channel[6] through a putative proton loading site (PLS) located between the heme $a_3$ propionates and a $Mg^{2+}$ center[7–10].

The oxygen reduction reaction catalyzed by C*c*O has been well-characterized[11–15]. As illustrated in Fig. 1B, the reaction is initiated by $O_2$ binding to the reduced enzyme, R, to generate the primary $O_2$-complex (A). Upon accepting 2 electrons and 2 substrate protons into the BNC, A is converted to the oxidized $O_H$ state, via the P and F intermediates. This oxidative phase of the reaction is followed by the reductive phase, where $O_H$ is reduced back to R via the $E_H$ intermediate, by accepting two additional electrons and 2 substrate protons into the BNC. During the reaction cycle, each time an electron and a substrate proton enter the BNC, a pumped proton is translocated across the mitochondrial membrane, as indicated by the white arrows.

[1]Department of Biochemistry, Albert Einstein College of Medicine, Bronx, NY 10461, USA. [2]Linac Coherent Light Source, SLAC National Accelerator Laboratory, Menlo Park, CA 94025, USA. [3]Department of Applied Physics, Stanford University, Stanford, CA 94305, USA. [4]Present address: Altos Labs, Redwood City, CA 94065, USA. ✉e-mail: syun-ru.yeh@einsteinmed.edu; denis.rousseau@einsteinmed.edu

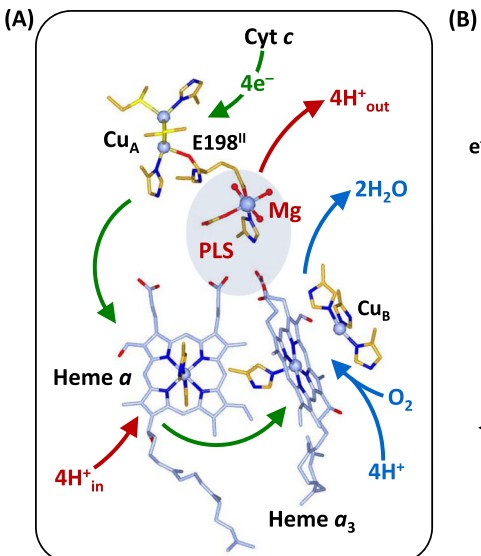

**Fig. 1 | Oxygen reduction reaction catalyzed by bCcO. A** Schematic illustration of the four redox active metal centers in bCcO and the electron and proton transfers associated with the O$_2$ reduction reaction. The entry of O$_2$ and four substrate protons into the heme $a_3$/Cu$_B$ binuclear center (BNC), as well as the release of the product water molecules out of it, are indicated by the blue arrows. The associated entry of four electrons into the BNC and the translocation of four pumped protons across the membrane are indicated by the green and red arrows, respectively. The putative proton loading site (PLS) between heme $a_3$ and the Mg center is highlighted by the light blue background. **B** The overall O$_2$ reduction reaction and the associated mechanism. The P intermediate is a general term for the P$_M$ and P$_R$ intermediates. The entry of the electrons and substrate protons into the BNC and the release of the product water molecules are indicated in each step of the reaction as described in the main text. The coupled proton translocation reactions are indicated by the white arrows. If the O$_H$ intermediate produced at the end of the oxidative phase is allowed to relax to the resting O state, its reduction to R does not support proton translocation.

Intriguingly, if the metastable O$_H$ state is allowed to relax to the resting O state, its reduction to R, unlike the O$_H$ →R transition, does not drive proton translocation[16,17], despite the fact that O and O$_H$ are redox equivalents.

The structural properties of O$_H$, distinguishing it from the O state, have been elusive. Time-resolved resonance Raman spectroscopic studies of bovine CcO (bCcO) revealed that the heme $a_3$ iron in the BNC of the O$_H$ state is coordinated by a hydroxide ion[12,18], as evidenced by its characteristic $\nu_{Fe\text{-}OH}$ stretching mode at 450 cm$^{-1}$. In contrast, a comparable $\nu_{Fe\text{-}OH}$ band has never been identified in the O state. As such, it has been thought that the inability of the O state to drive proton translocation is at least a partial result of a unique BNC ligation state[11,19,20]. The three dimensional structure of O$_H$ has not been determined, while the crystal structures of O have been reported for various homologs of CcO[21–23]. It is clear that, in the O state, strong electron density associated with exogeneous ligand(s) is present between heme $a_3$ and Cu$_B$ in the BNC; its assignment, however, has been controversial. It was first assigned to a peroxide ion bridging the two metal centers[21,24], but the best fitted O-O bond distance is much longer than that of a typical ferric peroxide species (~1.48 Å); in addition, it is unclear as to how the peroxides were formed and why they were stable under the equilibrium conditions[22]. As a result, alternative assignments have been considered. Based on theoretical perspectives, the BNC ligand has been proposed to be a dioxygen[25] or a water molecule[26]. In contrast, other crystallographic studies of bCcO (PDB ID: 7TIE)[27] and *R. sphaeroides* CcO (PDB ID: 2GSM)[22] suggest that a water and a hydroxide are coordinated to the heme $a_3$ and Cu$_B$, although the O-O distance between the two ligands (~1.9–2.0 Å) is too close for a typical H-bond.

Recently, it was recognized that macromolecular crystallographic structures obtained with intense synchrotron light sources at cryogenic temperatures often suffer from radiation damage problems, in particular for proteins containing redox sensitive metal centers[28–30]. It has been shown that, in the O structure of bCcO acquired with a typical synchrotron light source, all the four redox active metal centers were reduced, although its polypeptide scaffold remained in a native-like

conformation[27,31]. Accordingly, approaches, such as serial femtosecond X-ray crystallography (SFX)[32], have been developed and employed to overcome the radiation damage problems. With SFX, the diffraction patterns of randomly oriented microcrystals suspended in a solution jet are collected sequentially with an X-ray free electron laser (XFEL) before they are destroyed by the intense femtosecond laser pulses. As such, virtually radiation damage-free structures can be obtained at room temperature. Using SFX, Branden and coworkers obtained a radiation damage-free O structure of the $ba_3$ oxidase from *Thermus thermophilus*, based on which it was concluded that the electron density in the BNC was best modeled by a ligand with a single oxygen atom, either a water or a hydroxide ion[23]. Likewise, we have used SFX to determine the radiation damage free O structure of bCcO[33]; however, we found that it required two oxygen atoms to account for the ligand electron density in the BNC, although the resolution (2.9 Å) was insufficient to clarify the identity of the ligands. On the other hand, Hirata et al. employed a different point-by-point scanning approach, using an XFEL as a light source, to circumvent radiation damage problems, based on which the BNC ligands of the O derivative of bCcO were assigned as two co-existing peroxide moieties[34], implying possible heterogeneity of CcO samples[35,36]. More recently, Kolbe et al. used a cryo-EM technique to solve the structure of a bacterial CcO from *P. denitrificans*, which suggests that the BNC ligand is a peroxide with an O−O distance of 1.42 Å[37]. Despite these and other efforts, no consensus on the ligand identity has been reached to date.

Here, we sought to clarify the structural properties of the O derivatives of bCcO by using a combination of resonance Raman spectroscopy and SFX.

## Results
### Identification of the heme $a_3$ ligand by resonance Raman spectroscopy
To determine the identity of the heme $a_3$ iron ligand, we carried out resonance Raman spectroscopic studies in free soluton. The 413.1 nm

output from a krypton ion laser was selected as the excitation light source to selectively enhance the signals associated with heme $a_3$. We reasoned that if the heme $a_3$ iron ligand is a water or a hydroxide ion, it should be able to be exchanged with the solvent water molecules. Accordingly, we incubated the resting O derivative of bCcO in isotope-substituted $H_2^{18}O$ buffer for 12 h to ensure complete solvent exchange prior to the resonance Raman measurements. To prevent photo-reduction, the laser power was kept low (~5 mW) and the spectral acquisition time was kept short (3 min); in addition, the spectra from 6 fresh samples were acquired and summed to improve the signal-to-noise ratio of the spectrum. As a comparison, the spectrum of O in naturally abundant $H_2^{16}O$ buffer was obtained in the same fashion.

As shown in Fig. 2, an oxygen isotope sensitive band was identified at 451 cm⁻¹ in the $H_2^{16}O$ buffer, which shifted to 428 cm⁻¹ in the $H_2^{18}O$ buffer. The isotopic shift of 23 cm⁻¹ is consistent with the theoretical shift of a Fe-OH⁻ stretching mode ($\nu_{Fe-OH}$) (24 cm⁻¹), indicating that the heme $a_3$ iron ligand of the resting O state is a hydroxide ion. It should be noted that, this solvent isotope sensitive mode was not detected in prior studies[12], possibly due to the complications resulting from laser induced photodamage[38]. The $\nu_{Fe-OH}$ mode is identical to that found in the $O_H$ state[12,18], suggesting that the BNC ligands in O and $O_H$ are the same, contrary to the common belief that they are distinct[11,19,20].

It is noteworthy that this Fe-OH⁻ stretching frequency (451 cm⁻¹) is remarkably low as compared to those of other ferric heme species (~490–550 cm⁻¹)[39], indicating an unusually weak Fe-OH⁻ coordination bond. Hydroxide is generally a strong field ligand for ferric heme iron, which is associated with a strong Fe-OH⁻ bond and a low spin electronic configuration. However, previous resonance Raman[12] and electron paramagnetic resonance (EPR)[40] spectroscopic studies revealed that the heme $a_3$ in the O state, like that in the $O_H$ state, has an unusual high spin configuration. We attribute the weak Fe-OH⁻ bond and unique high spin configuration of the heme $a_3$ to a strong H-bond between the hydroxide ligand and its surrounding environment, as that detected in a hemoglobin from *M. tuberculosis*, which exhibits a similar low Fe-OH⁻ stretching frequency (454 cm⁻¹) and a high spin

electronic configuration due to a H-bond between the hydroxide ligand and a nearby tyrosine residue[41]. The presence of a strong H-bond to the hydroxide ligand in the BNC of bCcO is supported by the structural data discussed below. It is notable that previous computational studies[19] suggested that the $O_H$ state has a single hydroxide ligand bridging heme $a_3$ iron and $Cu_B$, thereby accounting for the high spin configuration of the heme $a_3$. However, if that were the case, the $\nu_{Fe-OH}$ mode of the $O_H$ state would be expected to be distinct from that of the O state due to the unique bonding interactions between the hydroxide ion and the two highly charged metal centers.

### Determination of the structure of the O state by SFX

For the SFX measurements, microcrystals of bCcO were prepared in the resting O state using a previously reported protocol[33]. To ensure the homogeneity of the O state, the microcrystals were reduced under anaerobic conditions and then allowed to turn over by exposing to $O_2$ and subsequently relax back to the O state by incubating overnight in a stabilizing solution. The suspension of the microcrystals was loaded into a gas-tight syringe and injected into the XFEL beam as a free solution jet with a single capillary MESH injector[42,43]. The XFEL beam, perpendicular to the solution jet, was directed into the tip of the Taylor cone formed at the output of the MESH injector. X-ray diffraction was collected for ~2 h, from which 84,736 indexable diffraction patterns were selected for structural analysis. The structure was solved and refined to a resolution of 2.38 Å (Supplementary Table 1). The structural markers for the oxidation states of the four redox centers confirm that the enzyme is in the fully oxidized O state (Supplementary Fig. 2).

### Structural characterization of the BNC

In the $F_O$-$F_C$ electron density map associated with the SFX data (Fig. 3A), a large 2-lobe electron density is evident in the BNC, indicating the presence of ligands coordinated to heme $a_3$ iron and $Cu_B$. As guided by the resonance Raman data (Fig. 2), we modeled the heme $a_3$ ligand density with a OH⁻ ion (Fig. 3B). In addition, we modeled the $Cu_B$ ligand density with a water molecule, as the coexistence of two negatively charged hydroxide ions in the BNC is expected to be energetically unfavorable. The occupancy of the ligands is confirmed by the polder maps shown in Fig. 3C. The Fe-OH⁻, $Cu_B$-$H_2O$, and Fe-$Cu_B$ bond lengths, which were unrestrained during the refinement, are determined to be 1.90, 2.14, and 4.74 Å, respectively. The O–O distance between the two oxygen ligands is 2.53 Å, which is shorter than a typical H-bond, but is consistent with a strong low-barrier H-bond[44,45]. This strong H-bond between the two ligands plausibly weakens the ligand field strength of the hydroxide and destabilizes the Fe-OH⁻ bond, thereby accounting for the high spin electronic configuration and the low Fe-OH⁻ stretching frequency revealed by the spectroscopic studies. Taken together our data demonstrate that the heme $a_3$ iron and $Cu_B$ in the O state, like those in the $O_H$ state, are coordinated by a hydroxide ion and a water molecule, respectively.

### Protonation state of Y244

$Cu_B$ in the BNC is coordinated by three histidine ligands, one of which (H240) forms a covalent bond with Y244 by a posttranslational modification (Fig. 4). It is well-established that, during the catalytic reaction, Y244 temporarily donates an electron and a proton to the heme $a_3$ bound $O_2$ to promote the O-O bond scission, by forming a tyrosyl radical[46], and that its re-reduction to a tyrosinate and re-protonation back to its neutral protonated form are tightly coupled to the ensuing electron and proton transfer processes[19,46,47].

Previously, with time-resolved SFX, we showed that, in a P intermediate of bCcO ($P_R$), Y244 is in the tyrosinate form[33]. Next to the Y244 there is a water molecule within H-bonding distance from the oxygen atom of the tyrosinate (indicated as $W^2$ in Fig. 4A), which is not present in the reduced R state (PDB IDs: 6NMF and 7THU) or the CO complex (PDB ID: 5W97, a structural analog of the **A** state), where Y244 is in the

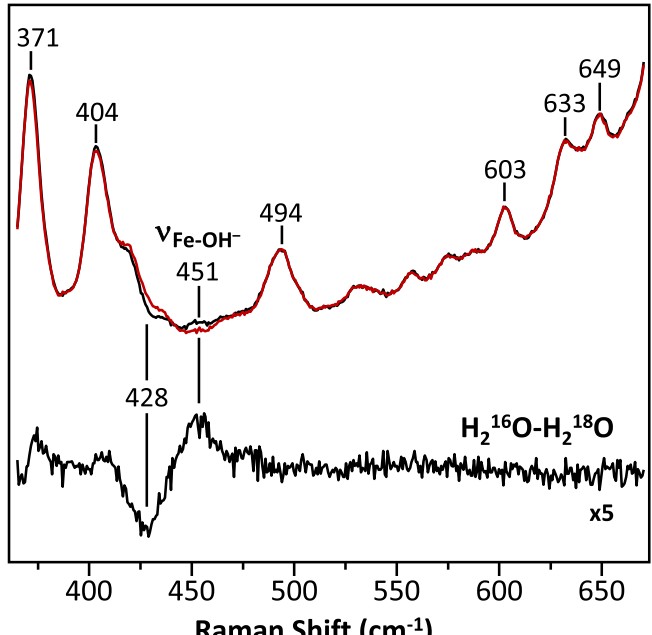

**Fig. 2 | Resonance Raman spectrum of the O state of bCcO in $H_2^{16}O$ (black) overlaid with that in $H_2^{18}O$ (red).** The $H_2^{16}O$-$H_2^{18}O$ difference spectrum (expanded by 5-fold) is shown in black at the bottom. The oxygen sensitive mode in the $H_2^{16}O$ sample centered at 451 cm⁻¹ that shifted to 428 cm⁻¹ in the $H_2^{18}O$ sample is assigned to the $\nu_{Fe-OH}$ mode.

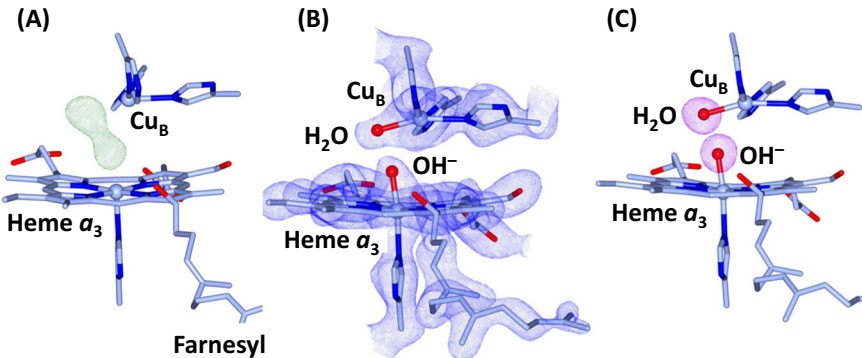

**Fig. 3 | Electron density maps of the BNC in the O state of bCcO. A** The $F_O$-$F_C$ electron density map (contoured at 7.0$\sigma$) showing clear 2-lobe electron density associated with the BNC ligands. **B** The $2F_O$-$F_C$ electron density map (contoured at 2.5$\sigma$) obtained with the electron density modeled with a hydroxide ion coordinated to the heme $a_3$ iron and a water molecule coordinated to Cu$_B$. **C** The Polder map (contoured at 7.0$\sigma$) associated with the BNC ligands.

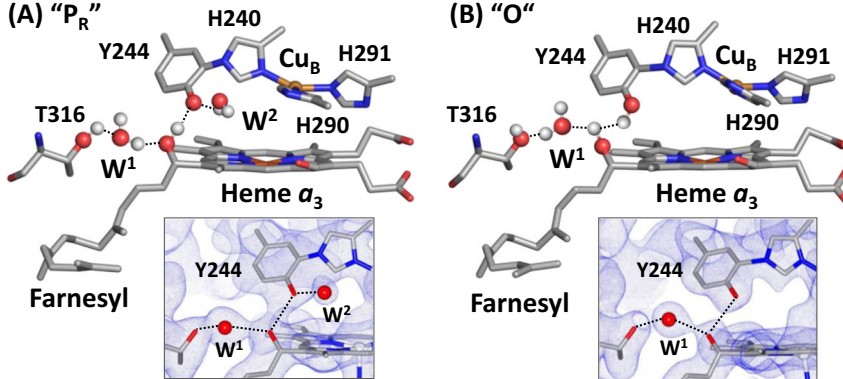

**Fig. 4 | Protonation state of Y244 in the $P_R$ (A) and O (B) states.** The post translationally modified Y244 forms a H-bonding network with the OH group of the farnesyl side chain of heme $a_3$, a water molecule (W$^1$) and T316. In the $P_R$ state, an additional water, W$^2$, is recruited into the heme $a_3$/Cu$_B$ binuclear center (BNC) to stabilize the tyrosinate configuration of Y244. This water is absent in the O state reported here, as evident in the $2F_O$-$F_C$ electron density map (contoured at 1.0$\sigma$) shown in the lower inset, signifying that Y244 is in the neutral protonated state. For clarity the BNC ligands are not shown. The oxygen atom and hydrogen atoms are shown as red and white spheres.

neutral protonated form. It suggests that, in the $P_R$ state, W$^2$ is recruited into the BNC to stabilize the tyrosinate configuration of Y244, similar to the water rearrangement induced by tyrosine deprotonation found in a photosynthetic reaction center from *B. viridis*[48]. Intriguingly, our current work reveals that W$^2$ is not present in the O state (Fig. 4B), signifying that Y244 is in the neutral protonated form, consistent with the conclusion drawn from computational studies carried out by Blomberg[49]. Although the structures of the F and $O_H$ intermediates have not been determined, infrared spectroscopic studies demonstrate that Y244 in both F and $O_H$, like that in the $P_R$ intermediate, is deprotonated[47], indicating that during the $P_R \rightarrow F \rightarrow O_H$ transition, Y244 remains in the tyrosinate configuration. This scenario is in good agreement with recent structural data showing the presence of W$^2$ in the mixed oxygen intermediates of bCcO[20,50]. Our current data further demonstrate that the $O_H \rightarrow O$ transition is associated with the protonation of the tyrosinate (Y244).

## Discussion

With the clarification of the O structure, we propose a complete reaction cycle of CcO as illustrated in Fig. 5. The reduced R state first binds $O_2$ to form the primary oxy intermediate A, with a $Fe^{3+}$-$O_2^-$ electronic configuration[51]. By accepting an electron and a proton from Y244, the O-O bond in A is heterolytically cleaved, leading to the formation of the putative $P_M$ intermediate, where one oxygen remains on the heme $a_3$ iron, in a ferryl ($Fe^{4+}$=$O^{2-}$) configuration, and the other oxygen is coordinated to Cu$_B$ as a hydroxide, while Y244 is converted

to a neutral tyrosyl radical. The entry of one electron into the BNC leads to the conversion of $P_M$ to $P_R$, where the tyrosyl radical is reduced to a tyrosinate. The $P_M \rightarrow P_R$ transition is associated with the entry of a new water (W$^2$, not shown in Fig. 5 for clarity), which stabilizes the tyrosinate configuration of Y244[33]. The subsequent entry of a substrate proton into the BNC further transforms $P_R$ to F, where the hydroxide ligand of Cu$_B$ is protonated to a water, which strengthens the iron-oxygen bond as evidenced by the shift of the frequency of the $Fe^{4+}$=$O^{2-}$ stretching mode from 785 to 804 cm$^{-1}$[52]. Finally, the entry of an additional electron and a substrate proton into the BNC converts F to $O_H$, where the $Fe^{4+}$=$O^{2-}$ moiety is reduced to the $Fe^{3+}$-$OH^-$ species[12,18,52]. During this oxidative phase, one pumped proton is translocated during each of the $P_R \rightarrow F$ and $F \rightarrow O_H$ transitions (as indicated by the white arrows) and, throughout the $P_R \rightarrow F \rightarrow O_H$ transformation, Y244 remains in the tyrosinate form.

In the ensuing reductive phase, an additional electron and a substrate proton enter the BNC, leading to the conversion of $O_H$ to $E_H$, where Cu$_B$ is reduced from the cupric to cuprous state and the hydroxide ligand of the heme $a_3$ is protonated to a water[11,53]. At the same time, the water ligand of Cu$_B$ is released out of the BNC. The entry of another electron and a substrate proton into the BNC leads to the conversion of $E_H$ to R, where the heme $a_3$ iron is reduced from the ferric to ferrous state and its water ligand is released out of the BNC; at the same time the tyrosinate is protonated back to its neutral form. It should be noted that in our proposed model the $E_H$ state has Y244 in the tyrosinate form and a water molecule coordinated to the heme $a_3$

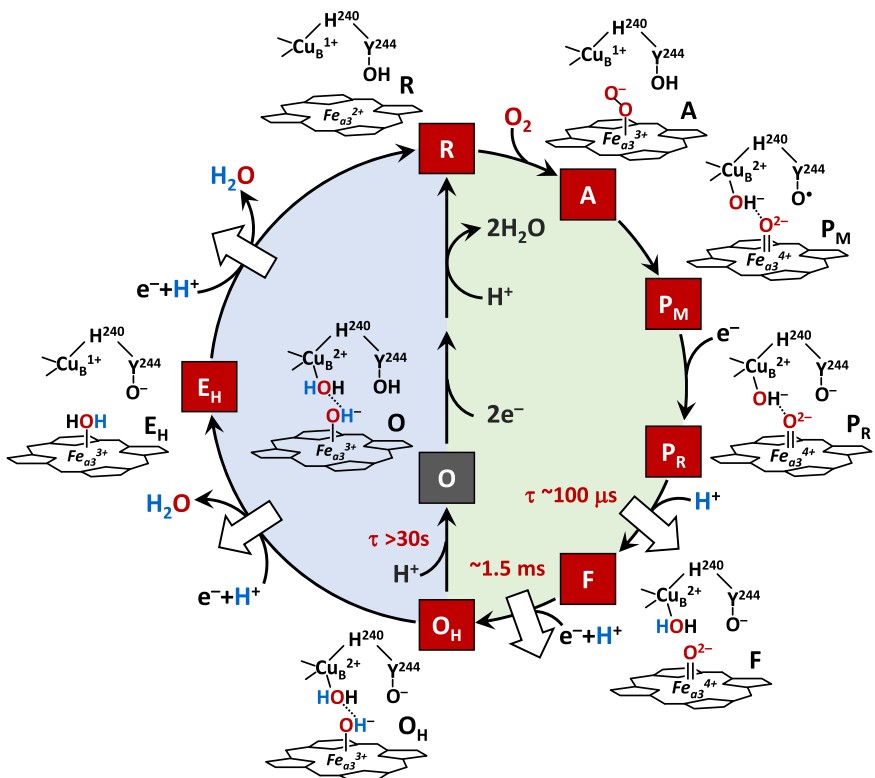

**Fig. 5 | Hypothesized ligand transformation in the BNC during the CcO reaction cycle.** The oxidative and reductive phase of the reaction cycle are highlighted with green and blue backgrounds, respectively. The white arrows indicate the proton translocation reactions associated with the reaction cycle. The lifetimes associated with the $P_R \rightarrow F$, $F \rightarrow O_H$ and $O_H \rightarrow O$ transitions are indicated in red.

iron[11,53], but an alternative configuration with Y244 in the protonated form and a hydroxide coordinated to the heme $a_3$ iron, as suggested by infrared spectroscopic studies[47], cannot be excluded. In any case, during this reductive phase, one additional pumped proton is translocated during each of the $O_H \rightarrow E_H$ and $E_H \rightarrow R$ transitions. Under electron deficient conditions, the $O_H$ intermediate produced at the end of the oxidative phase can spontaneously and slowly relax to the resting O state[16]. Our current data revealed that during the $O_H \rightarrow O$ transition, the tyrosinate (Y244) is protonated to the neutral form, leading to the dissociation of the $W^2$ that stabilizes the tyrosinate, while the BNC ligands remain unchanged.

The results reported here have important implications on proton translocation. It has been shown that the entry of the first two substrate protons into the BNC, associated with the $P_R \rightarrow F$ and $F \rightarrow O_H$ transitions, is mediated by the D-channel and occurs rapidly (within ~100 μs and 1.5 ms, respectively)[2]. Similarly, the entry of the other two substrate protons into the BNC, associated with the $O_H \rightarrow E_H$ and $E_H \rightarrow R$ transitions, also takes place quickly (on the submillisecond time scale), although it is mediated by a different channel, the K-channel[54]. In sharp contrast, the off-pathway $O_H \rightarrow O$ transition is sluggish and does not reach completion until ~30 s[16], suggesting that the tyrosinate (Y244) is likely protonated by an adventitious proton from its surrounding environment or some other part of the protein, rather than a substrate proton from the K-channel, despite the fact that Y244 sits at the end of the K-Channel (see Supplementary Fig. 1). If the proton were derived from the K-Channel, a much more efficient $O_H \rightarrow O$ transition and a unique sidechain conformation of K319 (a critical component of the K-Channel that is predicted to undergo conformational changes during proton translocation[55,56]), which was not detected in the current structure, would be expected.

Based on the electroneutrality principle proposed by refs. 57,58, during the catalytic reaction, the entry of each electron into the BNC

(see the green arrows in Fig. 6A) is charge-compensated by the entry of one substrate proton from the D or K-channel (blue arrows). The redox energy thereby derived is then used to drive the release of a pumped proton into the P-side of the membrane from the PLS, which is preloaded with the pumped proton(s) via the D or H-channel. As illustrated in Fig. 5, Y244 in all the intermediate states active in proton translocation ($P_R$, F, $O_H$, and $E_H$) is in the deprotonated tyrosinate form, suggesting that the premature protonation of the tyrosinate (Y244) during the $O_H \rightarrow O$ transition, without the input of any electron into the BNC, perturbs the charge balance in the BNC of the O state, thereby disabling the release of pumped protons out of the PLS upon reduction (Fig. 6B). This analysis of our structure is in good agreement with the conclusions drawn from recent computational studies carried out by Blomberg, showing that the protonation of Y244 associated with the $OH \rightarrow O$ transition lowers the reduction potential of the metal centers in the BNC, such that the proton translocation is no longer energetically favorable upon reduction[49]. These data demonstrate the importance of deprotonated forms of Y244 in modulating the energy landscape of the CcO reaction to promote the coupling of the oxygen reduction chemistry to proton translocation[53].

In summary, previous studies revealed that O and $O_H$ are redox equivalents, but only the latter, not the former, is capable of translocating protons upon reduction; with a combination of resonance Raman spectroscopy and SFX, we now clarify that the heme $a_3$ iron and $Cu_B$ in the BNC of the O state, like those in the $O_H$ state, are coordinated by a hydroxide ion and water, respectively; in addition, we show that the post-translationally modified Y244 is in the neutral protonated form, which distinguishes the O state from the $O_H$ state, where Y244 is in the deprotonated tyrosinate form. Our data underscore the pivotal role of the deprotonated form of Y244, a residue fully conserved in the CcO family of enzymes, in energizing the enzyme for proton translocation, as supported by reported computational studies[53].

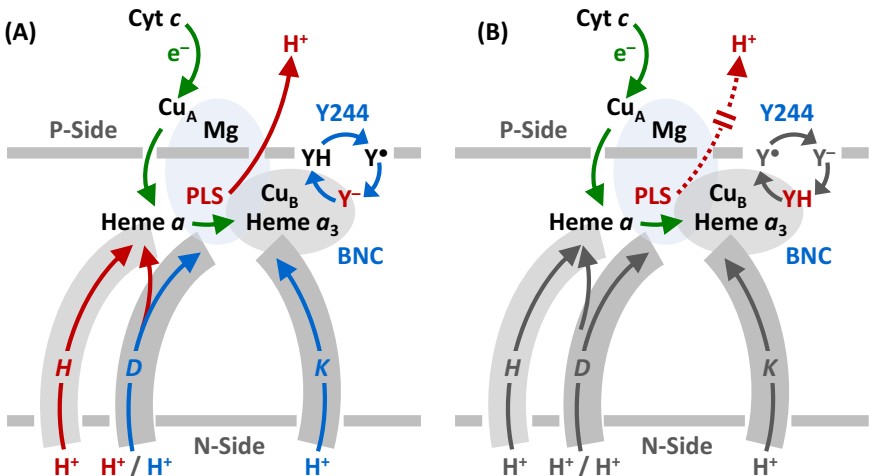

**Fig. 6 | Functional role of Y244 during the C*c*O reaction cycle. A** During the active turnover, Y244 cycles between the neutral form (YH), tyrosyl radical form (Y•) and deprotonated tyrosinate form (Y⁻) as depicted by the blue cycle. In all the intermediate states active in proton translocation, Y244 is in the Y⁻ form (in red), which ensures the tight coupling of electron transfer (green arrows) and substrate proton transfer (blue arrows), via the D and K-channel into the heme $a_3$/Cu$_B$ binuclear center (BNC), to drive the proton translocation (red arrows) via the D or H channel from the N to P-side of the membrane. **B** The protonation of the tyrosinate to the YH form (red) in the O state disables the proton translocation upon its reduction to R.

## Methods

### Protein sample preparation

Bovine C*c*O (bC*c*O) was isolated from bovine heart by slight modifications of a reported procedure[59,60] as described below. All the procedures were done at 4 °C. Bovine hearts, obtained from cows freshly slaughtered in a local slaughterhouse, were immediately cooled in a cold sodium phosphate buffer (pH 7.4, 50 mM). The skin, fat tissues and vascular tubes of the bovine hearts were trimmed from the heart and cut into small cubes. 550 g meat was grounded in a grinder. The ground meat was mixed with 3250 ml cold sodium phosphate buffer (pH 7.4, 23 mM) and homogenized in a blender at 13,000 rpm for 10 min. The solution was centrifuged at 2000 × *g* for 20 min. The supernatant was filtered through cheesecloth and kept aside. The precipitate was collected and mixed with 1500 ml cold sodium phosphate buffer (pH 7.4, 20 mM) and homogenized again in a blender at 13,000 rpm for 10 min. It was centrifuged at 2000 × g for 20 min and filtered through cheesecloth. The two red-colored supernatants were then combined and acidified to pH 5.15 with 30% acetic acid and centrifuged at 2000 × *g* for 15 min. The yellow soil-colored precipitate was collected and mixed with 400 ml cold water. It was then homogenized in a blender at 3500 rpm for 2 min. The volume was adjusted to 1.5 L with cold water and centrifuged at 2000 × *g* for 15 min. The tea-colored precipitate was collected and dissolved in 50 ml sodium phosphate buffer (pH 7.4, 200 mM). It was homogenized in a blender at 7000 rpm for 3 min and then centrifuged at 1500 × *g* for 10 min to remove the bubbles in the solution. The solution was mixed with 94 ml sodium phosphate buffer (pH 7.4, 200 mM) and the pH was adjusted to 7.4 with small aliquots of 3 N NaOH, and then water was added to make a final volume of 288 ml. The sample was kept at 4 °C overnight.

On the second day, 25 ml sodium cholate (40% w/v) was added to the sample, which was then subjected to seven rounds of ammonium sulfate fractionation at 4 °C. In the 1st ammonium sulfate fractionation, 61.35 g ammonium sulfate was added to the sample to achieve 33% saturation. The sample was stirred at 4 °C for 30 min and then centrifuged at 25,000 × *g* for 20 min. The supernatant was collected, and additional ammonium sulfate was added (with the pH kept at ~7.3–7.4 by adding small aliquots of 3 N NaOH) to achieve 50% saturation to precipitate out bC*c*O. The sample was centrifuged at 25,000 × *g* for 30 min. The precipitate was collected and homogenized in 0.5% (w/v) sodium cholate in sodium phosphate buffer (pH 7.4, 100 mM) with a final volume of 90 ml. The solution was dialyzed against 3 L sodium phosphate buffer (pH 7.4, 40 mM) for 90 min. It was then centrifuged at 100,000 × *g* for 50 min. The precipitate was collected and homogenized in 2.0% (w/v) sodium cholate in sodium phosphate buffer (pH 7.4, 100 mM) with a final volume of 100 ml. The sample was then subjected to the 2nd ammonium sulfate fractionation. Ammonium sulfate was added to the sample to achieve 25% saturation, which was then incubated at 4 °C for 30 min. It was centrifuged at 35,000 × *g* for 10 min. The supernatant was collected, and additional ammonium sulfate was added (with the pH kept at ~7.3–7.4) to achieve 45% saturation to precipitate out bC*c*O. The solution was centrifuged at 35,000 × *g* for 10 min. The precipitate was collected and homogenized in 0.5% (w/v) sodium cholate in sodium phosphate buffer (pH 7.4, 100 mM) with a final volume of 100 ml. In the 3rd ammonium sulfate fractionation, ammonium sulfate was added to the sample to reach 25% saturation. The solution was incubated at 4 °C for 30 min, and then centrifuged at 35,000 × *g* for 10 min. The supernatant was collected, and additional ammonium sulfate was added (with the pH kept at ~7.3–7.4) to achieve 40% saturation to precipitate out bC*c*O. The sample was centrifuged at 35,000 × *g* for 10 min. The precipitate was collected and homogenized in 0.5% (w/v) sodium cholate in sodium phosphate buffer (pH 7.4, 100 mM) with a final volume of 65 ml. In the 4th ammonium sulfate fractionation, ammonium sulfate was added to the sample to reach 25% saturation. It was incubated at 4 °C for 30 min, and then centrifuged at 35,000 × *g* for 5 min. The supernatant was collected, and additional ammonium sulfate was added (with the pH kept at ~7.3–7.4) to achieve 35% saturation to precipitate out bC*c*O. It was centrifuged at 35,000 × *g* for 5 min. The precipitate was collected and homogenized in 0.34% (w/v) n-Decyl-β-D-maltoside in sodium phosphate buffer (pH 7.4, 100 mM) with a final volume of 100 ml. In the 5th ammonium sulfate fractionation, ammonium sulfate was added to the sample to reach 40% saturation. The solution was centrifuged at 35,000 × *g* for 5 min. The supernatant was collected, and additional ammonium sulfate was added (with the pH kept at ~7.3–7.4) to achieve 60% saturation to precipitate out bC*c*O. It was centrifuged at 35,000 × *g* for 12 min. The precipitate was collected and homogenized in 0.2% (w/v) n-Decyl-β-D-maltoside in sodium phosphate buffer (pH 7.4, 100 mM) with a final volume of 50 ml. The solution was stored in a cold room overnight.

On the third day, the sample volume was increased to 110 ml by adding 0.2% (w/v) n-Decyl-β-D-maltoside in sodium phosphate buffer (pH 7.4, 100 mM). The sample was subjected to the 6th ammonium

sulfate fractionation. Ammonium sulfate was added to the sample to reach 50% saturation, which was then centrifuged at 35,000 × *g* for 5 min. The supernatant was collected, and additional ammonium sulfate was added (with the pH kept at ~7.3–7.4) to achieve 70% saturation to precipitate out bC*c*O. The sample was centrifuged at 35,000 × *g* for 15 min. The precipitate was collected and homogenized in 0.2% (w/v) n-Decyl-β-D-maltoside in sodium phosphate buffer (pH 7.4, 100 mM) with a final volume of 100 ml. In the last ammonium sulfate fractionation, ammonium sulfate was added to the sample to reach 55% saturation. The solution was centrifuged at 35,000 × *g* for 5 min. The supernatant was collected, and additional ammonium sulfate was added (with the pH kept at ~7.3–7.4) to achieve 70% saturation to precipitate out bC*c*O. The sample was centrifuged at 35,000 × *g* for 15 min. The precipitate was collected and dissolved in 5 ml sodium phosphate buffer (pH 7.4, 10 mM). It was dialyzed against 1 L fresh sodium phosphate buffer (pH 7.4, 10 mM) three times for 1 h, 2 h and 3 h each. After dialysis, the sample was centrifuged at 35,000 × *g* for 20 min. The supernatant was collected and concentrated to reach a protein concentration of ~80–90 mg /ml, using an Amicon Diaflo apparatus with an ultrafiltration membrane (Advantec) with a pore size of 200,000 Da.

On the fourth day, the concentrated sample was washed three times with 0.2% (w/v) n-Decyl-β-D-maltoside in sodium phosphate buffer (pH 7.4, 20 mM) using the Amicon Diaflo apparatus. It was followed by three additional washes with 0.2% (w/v) n-Decyl-β-D-maltoside in sodium phosphate buffer (pH 6.8, 40 mM). After the 2nd wash, the sample was further concentrated until micro-crystals were formed on top of the Amicon membrane. The micro-crystal slurry was collected and centrifuged at 35,000 × *g* for 10 min. The micro-crystal pellet was collected and dissolved in 0.2% (w/v) n-Decyl-β-D-maltoside in sodium phosphate buffer (pH 6.8, 40 mM) to generate the protein stock for the resonance Raman measurements and microcrystal preparation for the SFX studies. To determine the concentration of bC*c*O, the absorption spectra of the sample reduced by dithionite was acquired and the concentration was calculated based on the absorbance difference at 603 and 630 nm using an extinction coefficient, $\varepsilon^{red}$ (603–630 nm), of 46.6 nM$^{-1}$cm$^{-1}$. Typical yields of the purified enzyme are 150–330 mg.

It is well-known that oxidized bC*c*O can exist in two distinct forms, the so-called "fast" and "slow" forms[35,36], depending on the purification and preparation methods. The optical absorption spectra of the bC*c*O samples used in this work displayed a Soret maximum at 423 nm in both the solution phase and crystalline form (see Supplementary Fig. 3), indicating that they are the fast form (i.e., the active form) of the enzyme[35,36].

### Resonance Raman measurements

To prepare the sample in H$_2^{18}$O, a concentrated bC*c*O sample (in 40 mM pH 7.4 phosphate buffer containing 0.2% decylmaltoside) was diluted by tenfold with H$_2^{18}$O containing the same buffer and then incubated overnight prior to the resonance Raman measurements. The final protein concentration was 30 μM. An equivalent sample in H$_2^{16}$O was prepared in the same fashion as a comparison. Resonance Raman spectra were obtained by using 413.1 nm excitation from a Kr ion laser (Spectra Physics, Mountain View, CA). The laser beam was focused to a ~30 μm spot on a spinning sample cell. The scattered light, collected at right angle to the incident laser beam, was focused on the 100 μm wide entrance slit of a 1.25 m Spex spectrometer equipped with a 1200 grooves/mm grating (Horiba Jobin Yvon, Edison, NJ), where it was dispersed and then detected by a liquid-nitrogen cooled CCD detector (Princeton Instruments, Trenton, NJ). A holographic notch filter (Kaiser, Ann Arbor, MI) was used to remove the laser scattering. The Raman shift was calibrated by using indene (Sigma).

It has been shown that high laser power can induce photoreduction linked artifacts in the resonance Raman spectra of bC*c*O[38], which

prevented identification of the Fe-OH stretching mode in a prior study[12]. To avoid photo-induced artifacts, we kept the sample in a spinning quartz cell that rotates at 1000 rpm and kept the laser power low, ~5 mW at the output of the laser (prior to its passage through mirrors and a focusing lens to the sample). In addition, we limited the laser exposure time to 3 min for each sample. The final spectrum was obtained by averaging six spectra acquired from six individual fresh samples. Under these conditions none of the previously reported bands associated with photodamage derived from high laser power were found in the spectra. To confirm that there was no photoreduction, spectra of reduced bC*c*O were acquired under the same conditions. None of the reduced marker lines were detected in the spectra of the oxidized enzyme. To determine if there were any oxygen species containing oxygen atoms that were not exchangeable with H$_2^{18}$O, such as a peroxide, we carried out the following experiment. We first placed the enzyme in H$_2^{18}$O buffer to exchange all exchangeable oxygen to $^{18}$O. We then reduced the enzyme to the fully reduced state, to expel all oxygen ligands from the BNC, and exposed it to $^{18}$O$_2$ to initiate the turnover. Subsequently, we allowed the enzyme to relax back to the fully oxidized state and then acquired a spectrum. As a reference, we carried out a comparable reaction with H$_2^{16}$O and $^{16}$O$_2$ and acquired another spectrum. Our data showed that no isotope differences were detected other than that assigned as the Fe-OH stretching mode. To ensure that no artifacts present in the buffer background, spectra of the H$_2^{16}$O and H$_2^{18}$O buffers alone were obtained. It was confirmed that no isotopic differences in the 400–500 cm$^{-1}$ window were detected.

### Microcrystal preparation

The microcrystals were prepared with a previously reported method[33]. The crystal growth was initiated by mixing the protein stock with the precipitant solution (0.2% decylmaltoside and 2.5% PEG4000 in 40 mM pH 6.8 phosphate buffer) and a seeding solution (prepared by crushing and sonicating large crystals in the mother solution). The microcrystals were allowed to grow at 4 °C for ~36 h before they were harvested and characterized by polarized optical microscopy. The microcrystals have a planar shape with approximate dimensions of ~20 × 20 × 4 μm. To obtain a homogeneous sample of the oxidized **O** crystals for the SFX measurements, the microcrystals were reduced anaerobically by a minimum amount of dithionite in a glove box and then thoroughly washed with the mother solution to remove excess dithionite and its oxidized products. The reduced microcrystals were then exposed to O$_2$ to initiate the enzyme turnover. They were allowed to relax in a stabilizing solution (0.2% decylmaltoside and 6.25% PEG-4000 in 40 mM pH 6.8 sodium phosphate solution) at 4 oC for ~24 hours prior to the SFX measurements.

### SFX data acquisition and structural analysis

The SFX measurements were conducted at the Macromolecular Femtosecond Crystallography (MFX) end station of the Linac Coherent Light Source (LCLS) at the SLAC National Accelerator Laboratory. A bC*c*O microcrystal slurry was loaded into a gas tight syringe and driven by a HPLC pump into a 100 μm diameter capillary of a Microfluidic Electrokinetic Sample Holder (MESH) injector[42,43], which has recently been developed as a useful and reliable technique for delivering microcrystal slurries into the XFEL beam[61–65]. A high voltage (~+2500 V) was applied to the sample at the entrance of the capillary against a grounded waste collector. It was used to electro-focus the microcrystal jet down to a Taylor cone at the capillary output. The 10 keV/30 fs pulses from the X-ray free electron laser (XFEL) intersected the microcrystal slurry at the tip of the Taylor cone prior to its development into a thin jet, as the jet was too unstable for data collection, while the thick region of the cone gave too many multiple hits and a high background. The diameter of the XFEL beam was ~3 μm. The sample flow rate was set to ~3 μl/min and the sample temperature was kept at ~293 K. The XFEL wavelength was 1.24 Å. A series of diffraction

patterns from randomly oriented microcrystals were collected with a Rayonix MX340-XFEL CCD detector at a 30 Hz rep rate.

The quality and hit rate of the SFX data were monitored in real time using OM[66]. The data were collected for ~2 h. *Psocake*[67,68] was used to determine the initial diffraction geometry and find crystal hits. 84,736 patterns in the acquired dataset were indexed and merged with the *CrystFEL* program suite developed for SFX experiments[69,70]. The initial structure was solved with molecular replacement with Phaser-MR through the CCP4 program suite[71] using a 1.9 Å resolution structure of bC*c*O (PDB ID: 7TIE) as the search model. Waters and BNC ligands were excluded from the search model. Further model building was performed using Coot[72]. Structure refinements were done using Refmac5 and PDB-Redo[73]. The final structure was refined to a resolution of 2.38 Å (see Supplementary Table 1) (PDB ID: 8GCQ). Similar structures were observed in the two monomers of the bC*c*O dimer. All the structural data presented here are based on the first monomer (subunit A-M), as it exhibited a better resolution owing to crystal packing.

### Reporting summary

Further information on research design is available in the Nature Portfolio Reporting Summary linked to this article.

## Data availability

Atomic coordinates and structure factors of the O state of bCcO generated in this study have been deposited in the Protein Data Bank (PDB) under accession code 8GCQ. PDB codes of previously published structures used in this study are 7TIE, 6NMF, 7THU, 5W97 and 2GSM. Source data are provided with this paper. Any additional requests for data should be directed to the corresponding authors. Source data are provided with this paper.

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

## Acknowledgements

The SFX experiments were carried out at the LCLS at the SLAC National Accelerator Laboratory. LCLS is an Office of Science User Facility operated for the US Department of Energy Office of Science by Stanford University. Use of the LCLS, SLAC National Accelerator Laboratory, is supported by the U.S. Department of Energy, Office of Science and Office of Basic Energy Sciences under Contract No. DE-AC02-76SF00515. The HERA system for in helium experiments at MFX was developed by Bruce Doak and funded by the Max-Planck Institute for Medical Research. This work was supported by National Science Foundation (NSF) CHE–1404929 (D.L.R. and S.-R.Y.) and National Institutes of Health (NIH) grants S10 OD023453, P41 GM139687, GM126297 (D.L.R. and S.-R.Y.) and GM115773 (S.-R.Y.).

## Author contributions

S.R.Y., I.I., and D.L.R. designed experiments; I.I. isolated and crystallized bCcO and measured and interpreted the resonance Raman spectra; R.G.S. designed and operated the MESH injector; R.E.S. prepared the injector auxiliary equipment; F.P., S.L., B.H., F.R.M., and S.B. collected the SFX data; Z.S., A.P., C.W., and C.H.Y. processed the SFX data; S.R.Y., I.I., C.H.Y., and D.L.R. analyzed and interpreted the SFX data; S.R.Y. and D.L.R. drafted the manuscript and all authors contributed to the revisions.

## Competing interests

The authors declare no competing interests.
