## [Peer Review File · Nature Communications]

Reviewers' Comments:

Reviewer #1:

Remarks to the Author:

This work uses a combination of serial femtosecond X-ray crystallography and resonance Raman spectroscopy to assess the structure of cytochrome c oxidase, which is responsible for more than 90% of the oxygen consumption in biology. The focus is on the structure of the resting 'fully oxidized' form of this enzyme, which has in the past been quite controversial, probably mainly because of radiation damage in previous X-ray and cryo-EM studies.

This work is very solid and carefully conducted. It will be a classic in the research on cellular respiration and bioenergetics, and definitely deserves to be published in Nature Comm.

Yet, it is not perfect in its present form.

My major criticism concerns the authors' conclusions about the so-called Oh state, which was not studied here, but which the authors nevertheless think has the same ligand structure as that of the state O they are reporting on. The authors seem to neglect the computational work in refs. 20 and 53 that led to the proposal that the Oh state has but one oxygenous ligand, an OH⁻ that bridges high spin ferric iron and cupric copper, and excluding oxygenous ligands on both iron and copper. Their only experimental reason for this view is that the O state now studied has resonance Raman properties like those described earlier for state Oh. However, those properties are not exactly the same (could be normal variation?). However, the observed frequency of the Fe-OH stretch may not be much different for the case of a H-bond from water, or bonding the -OH to CuB.

So, I'd wish the authors would acknowledge the computational proposals of an OH⁻ bridge in the Oh state, since those studies did test the case with a water molecule bound to CuB.

Another comment concerns the conclusion that the tyrosine in the site is protonated in state O (relative to state Oh). This is an early proposal by M. Blomberg, whose computational work in this respect receives too little credit.

Yet another comment relates to the mechanism envisaged in Fig. 5. (and accompanying text).

The FTIR work in ref. 20 suggested that the tyrosine was protonated in state Eh along with reduction of CuB. The authors prefer protonation of the OH⁻ bound to heme iron - but without experimental support.

In 'Implications on proton translocation' the authors suggest that the tyrosinate is 'likely protonated by an adventitious proton...rather than by a proton from the K-channel'. This may not be the best way to describe it, because the most likely proton donor here would be the conserved lysine in the K-channel.

All in all, and summarising, this is excellent work on an extremely important biological reaction, work that will be cited decades from now. My comments above are just ideas to help improving the paper.

Reviewer #2:

Remarks to the Author:

In this article the authors present new data that distinguishes the O and O_H state of CcO by the protonation/deprotonation of the Y244 ligand. The measurements were performed by femtosecond x-ray crystallography with a free electron laser source (XFEL). Furthermore a hydroxide was identified as an axial ligand in the O state using resonance Raman spectroscopy. On this basis a possible explanation regarding the influence of Tyr244 on the proton pumping mechanism is given. The new data gives valuable insight in the still not understood reaction mechanism of CcO and is impressive. In my view, however, some crucial points were not discussed and some of the conclusions remain speculative. The manuscript therefore needs a

critical revision before it can be considered for publication in Nat. Commun. The main points are:

1. Recent publications on the topic that give a different (or supporting) interpretation on the axial ligand, the role of Tyrosine protonation and K-channel proton uptake were ignored (Nat. Commun. (2021) 12:6903, JACS (2021) 143, 2769, Chem. Sci. (2020) 11, 3804.). This is more explained in the points 3,4 and 5.

2. The Fe-OH stretching mode in the O state was identified via $H_{2^{16}O} - H_{2^{18}O}$ RR difference spectroscopy. It was noted in the article that these bands were not seen in other RR measurements before, most likely due to photodamage. However, the laser power and accumulation times used here (413 nm, 5 mW, 3 min) are actually quite high. It is therefore not clear how under these circumstances photoreduction is prevented.

3. Other publications (see above) have measured a peroxide as ligand in the O state. Also calculations do show stable configurations for both hydroxide and peroxide. This data cannot simply be dismissed as wrong. Most likely the different observations of a peroxide and a hydroxide ligand can be explained by heterogeneity of the samples. Such heterogeneity cannot be excluded from the present data as well since the very stable peroxide would not exchange with the $18O$ from water and therefore the RR difference spectra would exclusively show the portion of enzymes with a hydroxide ligand.

4. In the present work the O state is characterized by a protonated tyrosine. This assumption ignores accepted knowledge that the O state exists at least in two different conformations (named fast and slow form, Chem. Rev. 2015, 115, 1936) that, besides not pumping protons in the reductive phase, also show different catalytic reactivity. In a recent publication (JACS (2021) 143, 2769) the protonated tyrosine was experimentally observed only in the slow form of the enzyme. As the current measurements were carried out at pH 6.8, it is possible that also in the present work the slow form of the O state was monitored.

5. The most interesting interpretation of the data is the possible tuning of the proton pumping ability by Tyr244. As mentioned by the authors, the K-channel only provides substrate protons. The influence of Y244, situated in the K-channel, regarding proton pumping is not per se evident. As possible explanation it is stated in the text „that the protonation of Y244 associated with the OH→O transition lowers the reduction potential of the metal centers in the BNC, such that the proton translocation is no longer energetically favorable upon reduction.“ While this interpretation is interesting (and certainly possible), there is no experimental evidence shown in the manuscript that would support this assumption. If this is the case, said proton must be missing in the proton pumping pathway. Showing e.g. the deprotonation of possible proton loading sites such as the heme propionates would greatly improve the stated hypothesis.

6. The title is not very informative and should be made more concrete

Reviewer #3:

Remarks to the Author:

The manuscript "Structural basis for functional properties of cytochrome c oxidase" by Ishigami et al presents one SFX structure two Raman resonance spectroscopy experiments that define the structure of the ground state oxidized form of the enzyme. Together, the data support the hypothesis of an hydroxy molecule bound to the heme with a stabilizing water molecule and a neutral tyrosine at the active site.

Overall, the manuscript is well written, with a very clear background of the knowledge and current hypothesis to be investigated and supported with the data collected.

Major comments

- The background and current knowledge is split between the main introduction and the section on the oxygen cycle. These two parts should be combined at the beginning to show exactly what has been determined to date and distinguish the actual result. The authors did not discover what is described in all of the first and most of the second paragraph in the oxygen cycle section, obscuring that the one single result is that of determining the O species (and none of the other ones). Figure 5 could be combined with figure 1 and a single step scheme of the OH to O to R should be presented as the conclusion.

- Materials and methods. The SFX crystal preparation and stabilization of the O intermediate is described, which includes preparation of the samples in a glove box and handling under inert conditions. There is no such description for the preparation of the Raman measurements, so it is very unclear to me how the authors can claim that the spectra collected is of a single, O species, rather than a different species or a mixture.
- Raman spectroscopy. The difference spectra clearly shows a difference that can be explained from the isotopic ligand exchange. The authors mention that other Ferric species show stretches in the 490-550 cm⁻¹ region but do not define what the prominent stretch at 494 cm⁻¹ is.
- Structure solution. There need to be more details regarding the data for good evaluation of the structures. This includes details regarding the data handling post indexing and prior to phase calculation. Table 1 also requires at least R_{pim} (or equivalent). There is no indication of the resolution of the highest shell.
- Structure. The PDB report shows 2 copies of the the cCO in the asymmetric unit cell. The authors should supply a discussion regarding the resemblance of the chains (the figures show only one copy.. are they equivalent? the PDB report shows a high percentage of the chain having outliers, up to 30%, so better assurance of what the overall structure looks like would be important).
- Figure 3. The Fo-Fc map is contoured at 7.0 sigma, which is a very high contour level. What is the reason for this? is there other difference density in the area that is being cut? Typically fo-Fc maps are presented at 3 or 4 sigma and 2Fo-Fc maps at 1 (otherwise a total electron density per unit area should be quoted).
- Solution vs crystal spectroscopy. The Raman was collected in solution rather than micro or macrocrystals. How can the authors assure that the species are the same in both states? Since the structure solution is completely dependent on the Raman resonance spectroscopy, data should have been collected on the crystal structures and if possible under anaerobic conditions.
- photoreduction artifacts - The authors have not presented a power titration to show that at the radiation used chosen was correct to avoid these effects. This data is necessary when claiming that there is no photoreduction. or a comparison to a reduced spectrum.
- Extended data Fig. 2 - The authors use a comparison of their structure to two previously published structures in the reduced and oxidized forms respectively to determine that their structure is in the correct oxidation state. This argument is very weak, especially with the typical medium resolution achieved with these systems and crystals. Either the figure has to be great improved to indicate exactly which atoms we should be focusing on and how their change is associated with the changes in oxidation state. Electron density maps at the same contour level should be supplied as well (both 2Fo-Fc and Fo-Fc). Otherwise, spectroscopy (maybe UV-Vis) has to be performed in the crystal slurries to prove the oxidation state.

Minor comments

- Main, paragraph 4 - the XFEL structures are not radiation damage free. The sentence "As such, radiation damage-free structures" should be changed to "virtually radiation-damage free". According to some of the latest literature in the field, there are artifacts that can be seen in the femtosecond collected data, even if they do not influence the structures presented in this manuscript.
- Protonation state of Y244, paragraph 2 - Pr is not defined up to this point, only P.
- figure 2 - please change the color of the difference spectrum to differentiate it from the 16O data.
- Ext Fig 2 - This figure is very confusing, especially with the use of the same colors in the main structure as the reduced form. Atom labels need to be added and the specific residues of interest

clearly highlighted and justified as the changes are minor.

We thank the reviewers for their thoughtful and insightful comments. We have taken their comments seriously and have modified our manuscript as described below.

Reviewer #1

My major criticism concerns the authors' conclusions about the so-called Oh state, which was not studied here, but which the authors nevertheless think has the same ligand structure as that of the state O they are reporting on. The authors seem to neglect the computational work in refs. 20 and 53 that led to the proposal that the Oh state has but one oxygenous ligand, an OH⁻ that bridges high spin ferric iron and cupric copper, and excluding oxygenous ligands on both iron and copper. Their only experimental reason for this view is that the O state now studied has resonance Raman properties like those described earlier for state Oh. However, those properties are not exactly the same (could be normal variation?). However, the observed frequency of the Fe-OH stretch may not be much different for the case of a H-bond from water, or bonding the -OH to CuB. So, I'd wish the authors would acknowledge the computational proposals of an OH-bridge in the Oh state, since those studies did test the case with a water molecule bound to CuB.

We thank the reviewer for raising this question. Our current work is focused on the **O** state, hence there is not much we could say about the **OH** state without additional structural data. Nonetheless, the reviewer is correct that, based on the identical $\nu_{\text{Fe-OH}}$ modes detected in the Raman spectra, we propose that **O** and **OH** states have the same oxygen ligands. We believe that if it were a μ -hydroxo bridged Fe(III) and Cu(II), the $\nu_{\text{Fe-OH}}$ mode would appear to be very different due to the unique bonding interactions between the hydroxide ion and the two highly charged metal centers. This is now discussed in the revised manuscript.

Another comment concerns the conclusion that the tyrosine in the site is protonated in state O (relative to state Oh). This is an early proposal by M. Blomberg, whose computational work in this respect receives too little credit.

We apologize for the oversight. The earlier work by M. Blomberg is now cited in the revised manuscript.

Yet another comment relates to the mechanism envisaged in Fig. 5. (and accompanying text). The FTIR work in ref. 20 suggested that the tyrosine was protonated in state Eh along with reduction of CuB. The authors prefer protonation of the OH⁻ bound to heme iron - but without experimental support.

We fully agree with the reviewer that we could not exclude this alternate possibility. It is now discussed in the revised manuscript; in addition, new references are now added to support each of the two proposed configurations.

In 'Implications on proton translocation' the authors suggest that the tyrosinate is 'likely protonated by an adventitious proton...rather than by a proton from the K-channel'. This may not

be the best way to describe it, because the most likely proton donor here would be the conserved lysine in the K-channel.

We respectively disagree with the reviewer on this point. Y244 sits right at the end of the K-Channel (Supporting Fig. 1); If the proton were derived from the K-Channel, a much more efficient $\text{O}_\text{H} \rightarrow \text{O}$ transition and a unique sidechain conformation of K319, which was not detected in the current structure, would be expected. This is now discussed in the revised manuscript.

Reviewer #2

1. Recent publications on the topic that give a different (or supporting) interpretation on the axial ligand, the role of Tyrosine protonation and K-channel proton uptake were ignored (Nat. Commun. (2021) 12:6903, JACS (2021) 143, 2769, Chem. Sci. (2020) 11, 3804.). This is more explained in the points 3,4 and 5.

Please see the responses to points 3,4, 5 below.

2. The Fe-OH stretching mode in the O state was identified via H_2^{16}O minus H_2^{18}O RR difference spectroscopy. It was noted in the article that these bands were not seen in other RR measurements before, most likely due to photodamage. However, the laser power and accumulation times used here (413 nm, 5 mW, 3 min) are actually quite high. It is therefore not clear how under these circumstances photoreduction is prevented.

The laser power that we indicated (5 mW) was measured at the output of the laser (prior to its passage through mirrors and a focusing lens to the sample). Hence the actual power at the sample was much lower. In addition, the sample was kept in a rotating cell that rotates at 1000 rpm to minimize photo-induced damage. Furthermore, the final spectrum was obtained by averaging 6 spectra acquired from 6 individual fresh samples. Under these conditions, we did not detect any of the previously reported bands associated with photodamage derived from high laser power (JBC 264, 6604-6607, 1989). To confirm that there was no photoreduction, spectra of reduced bCcO were acquired under the same conditions. As shown below, none of the reduced marker lines were detected in the spectra of the oxidized enzyme. (It should be noted that the spectrum of the oxidized enzyme was obtained at the same conditions as that used to acquire the spectra shown in Fig. 2 except that it was integrated at a longer time, 10 min vs 3 min, to assess any possible buildup of a reduced form.) All the aforementioned control experiments are now discussed in the revised Material and Methods section.

3. Other publications (see above) have measured a peroxide as ligand in the O state. Also calculations do show stable configurations for both hydroxide and peroxide. This data cannot simply be dismissed as wrong. Most likely the different observations of a peroxide and a hydroxide ligand can be explained by heterogeneity of the samples. Such heterogeneity cannot be excluded from the present data as well since the very stable peroxide would not exchange with the ^{18}O from water and therefore the RR difference spectra would exclusively show the portion of enzymes with a hydroxide ligand.

We thank the reviewer for raising this question. The reviewer is correct that the bCcO sample might be a heterogeneous mixture of “fast” and “slow” forms of the enzyme depending on the purification and preparation methods. The optical absorption spectra of the bCcO samples used in this work displayed a Soret maximum at 423 nm in the solution phase, as well as in crystalline form, indicating that they are the fast form (i.e. the active form) of the enzyme. We have now discussed it and added references in the Material and Methods section. In addition, we have added Supplementary Fig. 3 to show the spectral data.

We have now discussed the cryoEM structural data reported by Kolbe *et al* in the 2021 Nat Comm paper in the Introduction section. We were aware that oxygen atoms in some oxygen species, such as peroxide, might not be exchangeable with H_2^{18}O ; hence we had carried out the following experiment. We first placed the enzyme in H_2^{18}O buffer to exchange all exchangeable oxygen to ^{18}O . We then reduced the enzyme to the fully reduced state, to expel all exogenous ligands from the BNC, and exposed it to $^{18}\text{O}_2$ to initiate the turnover. Subsequently, we allowed the enzyme to relax back to the fully oxidized state and then acquired a spectrum. As a reference, we carried out a comparable reaction with H_2^{16}O and $^{16}\text{O}_2$ and acquired another spectrum. Our data showed that no isotope differences were detected other than that assigned as the Fe-OH

stretching mode. This experiment is now described in the Materials and Methods section of the revised manuscript.

4. In the present work the O state is characterized by a protonated tyrosine. This assumption ignores accepted knowledge that the O state exists at least in two different conformations (named fast and slow form, Chem. Rev. 2015, 115, 1936) that, besides not pumping protons in the reductive phase, also show different catalytic reactivity. In a recent publication (JACS (2021) 143, 2769) the protonated tyrosine was experimentally observed only in the slow form of the enzyme. As the current measurements were carried out at pH 6.8, it is possible that also in the present work the slow form of the O state was monitored.

The nice experiments reported in the JACS paper were done on the bacterial enzyme from *R. sphaeroides*. The fast/slow transition could have a different effect on the tyrosine configuration in the bacterial *versus* bovine enzyme. The Soret maximum of our enzyme in both solution phase and crystalline state is at 423 nm, based on which we are confident that we had the fast form (active form) of the enzyme. We have now discussed it and added new references (including the JACS paper suggested by the reviewer) in the Material and Methods section. In addition, we have added Supplementary Fig. 3 to show the spectral data.

5. The most interesting interpretation of the data is the possible tuning of the proton pumping ability by Tyr244. As mentioned by the authors, the K-channel only provides substrate protons. The influence of Y244, situated in the K-channel, regarding proton pumping is not per se evident. As possible explanation it is stated in the text „that the protonation of Y244 associated with the OH→O transition lowers the reduction potential of the metal centers in the BNC, such that the proton translocation is no longer energetically favorable upon reduction.“ While this interpretation is interesting (and certainly possible), there is no experimental evidence shown in the manuscript that would support this assumption. If this is the case, said proton must be missing in the proton pumping pathway. Showing e.g. the deprotonation of possible proton loading sites such as the heme propionates would greatly improve the stated hypothesis.

We agree with the reviewer that there is a possibility that proton is depleted in the PLS in the O-state. Unfortunately, with the current structural data, we were unable to determine the protonation states of the critical residues/moieties in the proton pumping pathways or PLS (such as the heme propionates). Apparently, additional experimental and computational studies will be required to further evaluate this model.

6. The title is not very informative and should be made more concrete

The reviewer is correct that our old title was too vague. We have now modified the title to “Structural insights into functional properties of the oxidized form of cytochrome *c* oxidase” to specifically indicate that the target of our study is the oxidized form of the enzyme.

Reviewer #3

- The background and current knowledge is split between the main introduction and the section on the oxygen cycle. These two parts should be combined at the beginning to show exactly what has been determined to date and distinguish the actual result. The authors did not discover what is described in all of the first and most of the second paragraph in the oxygen cycle section, obscuring that the one single result is that of determining the O species (and none of the other ones). Figure 5 could be combined with figure 1 and a single step scheme of the OH to O to R should be presented as the conclusion.

All the past discoveries made by us or others described in our manuscript were clearly cited; hence there should be no confusion. Nonetheless, we have carefully considered the reorganization suggested by the reviewer. However, we found that the current organization is best for general readers to comprehend the complex story and to appreciate the conclusion drawn from this work.

- Materials and methods. The SFX crystal preparation and stabilization of the O intermediate is described, which includes preparation of the samples in a glove box and handling under inert conditions. There is no such description for the preparation of the Raman measurements, so it is very unclear to me how the authors can claim that the spectra collected is of a single, O species, rather than a different species or a mixture.

We thank the reviewer for raising this question. It is well-known that oxidized bCcO can exist in two distinct forms, the so-called “fast” and “slow” forms, depending on the purification and preparation methods. The optical absorption spectra of the bCcO samples used in this work displayed a Soret maximum at 423 nm in the solution phase, as well as in crystalline form, indicating that they are the fast form (i.e. the active form) of the enzyme. We have now discussed it and added references in the Material and Methods section. In addition, we have added Supplementary Fig. 3 to show the spectral data.

- Raman spectroscopy. The difference spectra clearly shows a difference that can be explained from the isotopic ligand exchange. The authors mention that other Ferric species show stretches in the 490-550 cm^{-1} region but do not define what the prominent stretch at 494 cm^{-1} is.

The 494 cm^{-1} line is a heme vibrational mode, not an oxygen-related mode, as it was not shifted in the H_2^{16}O - H_2^{18}O isotope difference spectrum (Fig. 2). To confirm that there were no artifact bands derived from photodamage to the bCcO sample, we conducted several control experiments, which are now discussed in a new paragraph in the revised Material and Methods section.

- Structure solution. There need to be more details regarding the data for good evaluation of the structures. This includes details regarding the data handling post indexing and prior to phase calculation. Table 1 also requires at least Rpim (or equivalent). There is no indication of the resolution of the highest shell.

The workflow associated with SFX data analysis has been well-developed and routinely used for structural determination of macromolecules. We have now clearly stated in the Material and Methods section that the acquired dataset was indexed with the *CrystFEL* program suite developed for SFX experiments and that a total of 84,736 patterns were indexed and then integrated and merged, before they were used to determine the initial structure with molecular replacement.

We have added the values associated with the highest resolution shell and R_{split} (equivalent to R_{pim}) in Supporting Table 1.

- Structure. The PDB report shows 2 copies of the the cCO in the asymmetric unit cell. The authors should supply a discussion regarding the resemblance of the chains (the figures show only one copy.. are they equivalent? the PDB report shows a high percentage of the chain having outliers, up to 30%, so better assurance of what the overall structure looks like would be important).

The reviewer is correct that there are two copies of bCcO in the asymmetric unit as bCcO exists as a dimer in the crystalline state. Similar structures were observed in the two monomers of the dimer. All the structural data presented here are based on the first monomer (subunit A-M), as it exhibits a better resolution owing to crystal packing. This is now clearly stated in the Material and Methods section.

- Figure 3. The Fo-Fc map is contoured at 7.0 sigma, which is a very high contour level. What is the reason for this? is there other difference density in the area that is being cut? Typically fo-Fc maps are presented at 3 or 4 sigma and 2Fo-Fc maps at 1 (otherwise a total electron density per unit area should be quoted).

The ligand density, as validated by the polder map shown in Fig. 3C, was very strong, hence the Fo-Fc map was contoured at 7 sigma, instead of 3 or 4 sigma. There was no difference density detected other than those associated with the ligand density.

- Solution vs crystal spectroscopy. The Raman was collected in solution rather than micro or macrocrystals. How can the authors assure that the species are the same in both states? Since the structure solution is completely dependent on the Raman resonance spectroscopy, data should have been collected on the crystal structures and if possible under anaerobic conditions.

To acquire resonance Raman spectra in solution phase, we need to rotate the sample at 1000 rpm to avoid photo-induced damage to the samples, which can't be done with microcrystal samples due to light scattering problems. Nonetheless, we did obtain the optical absorption spectra of the enzyme in solution phase and crystalline state, which showed the same Soret maximum at 423 nm, demonstrating that they are in the same active form. This information is now clearly stated in the Material and Methods section, and the spectral data are now presented in Supplementary Fig. 3.

- photoreduction artifacts - The authors have not presented a power titration to show that at the radiation used chosen was correct to avoid these effects. This data is necessary when claiming that there is no photoreduction. or a comparison to a reduced spectrum.

Please see response #2 to the reviewer #2.

- Extended data Fig. 2 - The authors use a comparison of their structure to two previously published structures in the reduced and oxidized forms respectively to determine that their structure is in the correct oxidation state. This argument is very weak, especially with the typical medium resolution achieved with these systems and crystals. Either the figure has to be great improved to indicate exactly which atoms we should be focusing on and how their change is associated with the changes in oxidation state. Electron density maps at the same contour level should be supplied as well (both $2F_o-F_c$ and F_o-F_c). Otherwise, spectroscopy (maybe UV-Vis) has to be performed in the crystal slurries to prove the oxidation state.

We agree with the reviewer that this figure was poorly presented. We have revised the figure by superimposing the structures of O* and R to highlight the structural differences between the two states in the redox-sensitive structural regions. In addition, the $2F_o-F_c$ electron density maps associated with the O state (contoured at 1.5σ) are now shown to justify the assignment of the structure.

The optical absorption spectra of the enzyme samples used in this work (in solution phase and crystalline state) are now presented in Supplementary Fig. 3.

Minor comments

- Main, paragraph 4 - the XFEL structures are not radiation damage free. The sentence "As such, radiation damage-free structures" should be changed to "virtually radiation-damage free". According to some of the latest literature in the field, there are artifacts that can be seen in the femtosecond collected data, even if they do not influence the structures presented in this manuscript.

We have made this change as suggested.

- Protonation state of Y244, paragraph 2 - Pr is not defined up to this point, only P.

P_R was defined earlier in Fig. 1 caption in the original manuscript. To make it clearer we have now revised the text to clarify it.

- figure 2 - please change the color of the difference spectrum to differentiate it from the ^{16}O data.

We have modified the figure caption to make it clear that the trace on the bottom is a difference spectrum and not the ^{16}O trace.

- Ext Fig 2 - This figure is very confusing, especially with the use of the same colors in the main structure as the reduced form. Atom labels need to be added and the specific residues of interest clearly highlighted and justified as the changes are minor.

This figure has been totally revised as discussed above.

Reviewers' Comments:

Reviewer #1:

Remarks to the Author:

The authors have adequately responded to my criticism.

Reviewer #2:

Remarks to the Author:

The questions raised in my first review have been adequately addressed. I would have liked to see more experimental evidence for the coupling between Y244 protonation and the proton pumping mechanism but I accept that it would go beyond the scope of this manuscript. I support publication in the current form.

Reviewer #3:

Remarks to the Author:

The authors addressed the comments well. My questions were answered and the manuscript revised accordingly, with the necessary information added and figures re-made.